# ContinualCropBank: Object-Level Replay for Semi-Supervised Online Continual Object Detection

## Abstract

Deep learning has achieved remarkable progress in object detection, but most advances rely on static, fully labeled datasets—an unrealistic assumption in dynamic, real-world environments. Continual Learning (CL) aims to overcome this limitation by enabling models to acquire new knowledge without forgetting prior tasks; however, many approaches assume known task boundaries and require multiple passes over the data. Online Continual Learning (OCL) offers a more practical alternative by processing data in a single pass; however, it remains limited by its dependence on costly annotations. To address this limitation, Label-Efficient Online Continual Object Detection (LEOCOD) extends OCL with a semi-supervised formulation, enabling detectors to leverage unlabeled data alongside limited labeled samples. In this paper, we propose *ContinualCropBank*, an object-level replay module for LEOCOD that stores object patches cropped from bounding box regions and pastes them into stream images during training. This solution enables fine-grained replay, mitigating catastrophic forgetting while addressing foreground–background imbalance and the scarcity of small objects. Experiments on two benchmark datasets demonstrate that incorporating ContinualCropBank improves detection accuracy and resilience to forgetting, achieving gains of up to 9.57 percentage points in average accuracy and reducing degradation from forgetting by up to 2.32 points.

## 1 Introduction

Recent advances in deep learning have dramatically improved performance on challenging computer vision tasks such as object detection (Zou et al., 2023). The widespread availability of powerful Graphics Processing Units (GPUs) and large-scale labeled datasets (Everingham et al., 2010; Lin et al., 2014) has enabled the training and rigorous evaluation of increasingly sophisticated neural network architectures. Consequently, real-world applications of computer vision in diverse domains such as autonomous driving (Grigorescu et al., 2020), healthcare (Esteva et al., 2019), and remote sensing (Zhu et al., 2017) have seen substantial gains. However, these achievements rely on offline learning with static, fully curated datasets, whereas many real-world environments demand models capable of updating continuously as new data becomes available.

The offline learning paradigm assumes full access to the entire dataset in advance—an assumption rarely satisfied in real-world scenarios. In practice, data often arrive sequentially, requiring continual model updates and motivating research in Continual Learning (CL) (De Lange et al., 2022). In this paradigm, the model is exposed to a sequence of tasks that arrive incrementally over time, often with limited or no access to data from previous tasks. A major challenge in this setting is catastrophic forgetting (McCloskey & Cohen, 1989), where learning new tasks leads to performance degradation on earlier ones. Moreover, CL methods assume that tasks arrive separately in batches with a predefined order, allowing multiple iterations over these batches. Since this assumption is unrealistic, Online Continual Learning (OCL) (Aljundi et al., 2019) addresses this limitation by eliminating artificial task boundaries and processing a data stream once per sample, making forgetting emerge naturally.

However, OCL requires annotations to be available almost immediately after samples are observed. This dependency significantly limits the applicability of OCL in many real-world scenarios. For in-

stance, continuous manual labeling is impractical in Earth-observing nanosatellites (Lucia et al., 2021) and remote wildlife monitoring (Balakrishnan et al., 2025) since reliable communication cannot be guaranteed. To reduce label demand, Wu et al. (2023) formalized Label-Efficient Online Continual Object Detection (LEOCOD), combining OCL with a semi-supervised formulation. In LEOCOD, each mini-batch contains video frames partitioned into a fixed proportion of fully labeled and unlabeled images. This formulation is similar to Semi-Supervised Object Detection (SSOD) (Liu et al., 2021), where models are trained on a mix of labeled and unlabeled images; however, SSOD assumes an offline training setting with multiple passes over the dataset. Consequently, a natural research pathway is to adapt SSOD methods to the LEOCOD context. Efficient-CLS (Wu et al., 2023) follows this direction by integrating the teacher-student mutual learning strategy (Liu et al., 2021) and an episodic memory for experience replay. Despite the accuracy gains of SSOD methods, their role in mitigating forgetting under LEOCOD remains underexplored.

This paper introduces *ContinualCropBank* (Figure 1a), a novel object memory module inspired by CropBank (Zhang et al., 2022), initially developed for SSOD. ContinualCropBank stores object patches cropped from the bounding-box regions of labeled images. During training, these stored patches are pasted into labeled and unlabeled stream images, enabling fine-grained replay that mitigates catastrophic forgetting. Unlike CropBank, which assumes unrestricted memory capacity, ContinualCropBank enforces per-category limits to remain practical for OCL scenarios. Furthermore, ContinualCropBank directly addresses two common challenges in SSOD: foreground-imbalance and the scarcity of small objects. To this end, stored patches are rescaled to generate additional examples of small objects and pasted into stream images to increase the proportion of foreground regions. To the best of our knowledge, ContinualCropBank is the first object-level replay module explicitly designed for LEOCOD, combining bounded per-class memory with scale-aware augmentation to mitigate forgetting while improving detection under limited supervision.

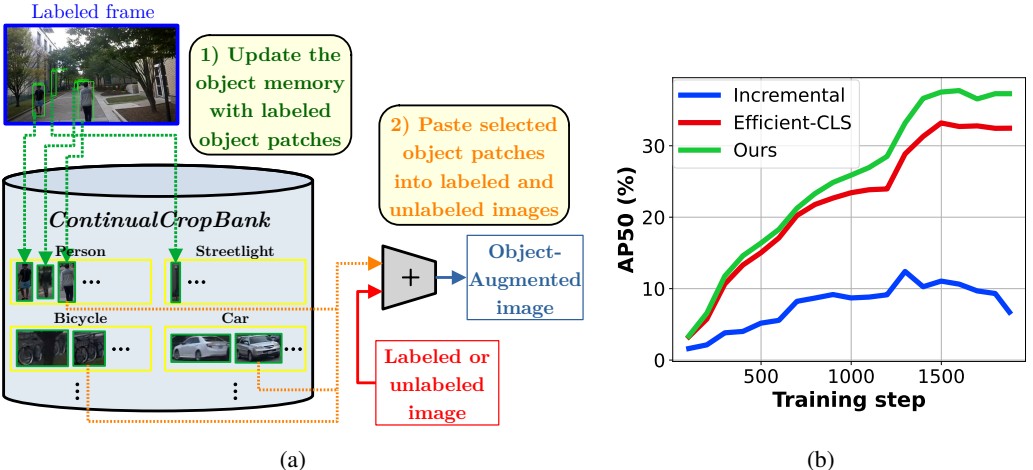

(a)                                    (b)

Figure 1: **(a)** The ContinualCropBank is an object memory that stores a fixed number of object regions cropped from labeled images, updated as new data arrive. Object patches are sampled from this memory and pasted into labeled and unlabeled stream images, producing object-augmented images for training. Image frame from the OAK dataset (Wang et al., 2021). **(b)** Detection performance of LEOCOD methods for the OAK dataset with $12.5\%$ annotation cost. Compared to Efficient-CLS, ContinualCropBank achieves consistent gains in detection performance over training steps.

We evaluate the effectiveness of ContinualCropBank using two benchmark datasets for online continual object detection, consisting of annotated image frames extracted from video streams and organized in temporal order (Wang et al., 2021; Zhu et al., 2023). We compare our method against state-of-the-art and baseline approaches under two experimental settings: fully supervised and semi-supervised, with varying proportions of labeled images per mini-batch. Our results show that integrating ContinualCropBank into the Efficient-CLS framework consistently improves detection accuracy and reduces forgetting compared to the state-of-the-art method (Figure 1b). For example, ContinualCropBank achieves gains of up to $9.57$ points in average detection performance and reduces degradation due to forgetting by up to $2.32$ points. Our main contributions are threefold:

1. We propose ContinualCropBank, an object-level replay module for LEOCOD that augments stream images with stored object patches, enabling fine-grained experience replay to mitigate catastrophic forgetting.

2. We leverage ContinualCropBank to address two well-known SSOD challenges within the CL setting: foreground–background imbalance and the scarcity of small objects.

3. We provide extensive empirical evidence that ContinualCropBank alleviates catastrophic forgetting and improves detection performance, surpassing state-of-the-art methods across multiple experimental settings.

## 2 RELATED WORK

**Continual Learning.** Continual Learning (CL) requires models to learn new tasks sequentially while preserving performance on previously learned ones. A common scenario is the class-incremental learning (Masana et al., 2023), where new classes appear in successive stages. The main challenge is to prevent catastrophic forgetting (McCloskey & Cohen, 1989) since past data cannot be stored or revisited due to memory or privacy constraints. Rehearsal-based methods address this challenge by replaying a small memory of past examples. For instance, Incremental Classifier and Representation Learning (iCaRL) (Rebuffi et al., 2017) replays a limited set of exemplar images from previous classes during training to reinforce existing knowledge. Greedy Sampler and Dumb Learner (GDumb) (Prabhu et al., 2020) retrains from scratch on a small, continuously updated memory buffer. Gradient Episodic Memory (GEM) (Lopez-Paz & Ranzato, 2017) constraints new gradient updates using limited stored samples from previous tasks. Averaged GEM (A-GEM) (Chaudhry et al., 2019) enforces constraints on average gradient directions rather than individual tasks. Some methods originally proposed for task-incremental CL, such as iCaRL and GDumb, have also been shown to be effective in Online Continual Learning (OCL) (Aljundi et al., 2019).

**Continual Object Detection.** Most CL methods for object detection focus on the class-incremental setting (Menezes et al., 2023), using strategies such as knowledge distillation losses (Shmelkov et al., 2017; Peng et al., 2020), meta-learning (Joseph et al., 2022), and replay-based approaches (Acharya et al., 2020). These methods typically adapt object detection benchmarks by splitting annotations into subsets of object categories that are revealed incrementally. In contrast, OCL methods for object detection are often evaluated on temporally ordered datasets derived from video streams. For example, Wang et al. (2021) introduce the Objects Around Krishna (OAK) benchmark and evaluate iCaRL and EWC in this context. Wagner et al. (2025) propose Configurable Recall, which maintains a memory buffer of previously seen data to perform periodic recalls of different sizes and frequencies. More recently, Wu et al. (2023) proposed Efficient-CLS, which combines a teacher-student paradigm (Liu et al., 2021) with experience replay, allowing for learning from both labeled and unlabeled data while alleviating catastrophic forgetting.

**Semi-Supervised Object Detection.** Semi-Supervised Object Detection (SSOD) leverages a small labeled set and a large pool of unlabeled images to improve detection performance. A dominant approach is the teacher-student mutual learning strategy (Liu et al., 2021), inspired by Mean Teacher (Tarvainen & Valpola, 2017) and FixMatch (Sohn et al., 2020). Several methods extend this framework to address specific challenges related to SSOD. For example, Mixed Pseudo Labels (MixPL) (Chen et al., 2023) augments images to balance object sizes and increase object density. Adaptive Class-Rebalancing Self-Training (ACRST) (Zhang et al., 2022) introduces an object memory bank that stores labeled objects, allowing instance-level copy-and-paste augmentation and mitigating class imbalance. Furthermore, SSOD methods are typically evaluated in an offline setting, with multiple passes over the dataset; thus, catastrophic forgetting is not a central concern. While teacher-student paradigm has been adapted to the LEOCOD scope (Wu et al., 2023), other SSOD advances remain largely unexplored in CL contexts.

## 3 METHOD

### 3.1 PROBLEM DEFINITION

This paper addresses the task of OCL for object detection in a semi-supervised setting. Wu et al. (2023) formalize this problem as Label-Efficient Online Continual Object Detection (LEOCOD). In

this setting, the model trains on a temporal sequence of mini-batches $\{D_1, \ldots, D_M\}$, where each $D_t$ consists of frames extracted from video streams. Within each mini-batch $D_t = D_t^l \cup D_t^u$, a fixed proportion of images are fully labeled, denoted as $D_t^l = \{(\mathbf{x}_i^l, \mathbf{y}_i^l)\}_{i=1}^{N_l}$, where $\mathbf{x}_i^l$ represents a labeled image, and $\mathbf{y}_i^l$ contains the corresponding object annotations (bounding boxes and category labels for all objects in the image). The remaining images are unlabeled, denoted as $D_t^u = \{(\mathbf{x}_i^u)\}_{i=1}^{N_u}$, and contain no associated annotations. After the algorithm processes a mini-batch, it is immediately discarded, and training proceeds to the subsequent mini-batch in the sequence. This one-pass constraint reflects the realistic limitations of OCL, where storage and revisit of past data are not feasible.

## 3.2 OVERVIEW OF EFFICIENT-CLS

Efficient-CLS (Wu et al., 2023) tackles the LEOCOD problem by incorporating the teacher-student strategy (Liu et al., 2021), originally developed for SSOD. In this framework, the teacher model (slow learner) generates predictions for unlabeled images $\mathbf{x}_i^u$, retaining those with confidence scores higher than a predefined threshold $\tau$ as pseudo-labels $\mathbf{y}_i^{pl}$. The unlabeled data are then augmented (*e.g.*, flipping, rotating, and scaling) and their augmented versions $(\tilde{\mathbf{x}}_i^u, \tilde{\mathbf{y}}_i^{pl})$ are combined with the labeled examples $\{(\mathbf{x}_i^l, \mathbf{y}_i^l)\}_{i=1}^{N_l} \cup \{(\tilde{\mathbf{x}}_i^u, \tilde{\mathbf{y}}_i^{pl})\}_{i=1}^{N_u}$ to train the student model (fast learner) using supervised learning. The overall training loss $\mathcal{L}$ for the student model consists of a supervised component $\mathcal{L}_{sup}$ for the labeled data and an unsupervised component $\mathcal{L}_{unsup}$ for the pseudo-labeled data:

$$\mathcal{L} = \mathcal{L}_{sup} + \lambda \mathcal{L}_{unsup}, \tag{1}$$

where $\lambda$ controls the contribution of the unlabeled samples to the total loss. The supervised loss $\mathcal{L}_{sup}$ includes both classification and regression losses for the Region Proposal Network (RPN) and Region of Interest (RoI) head, while the unsupervised loss $\mathcal{L}_{unsup}$ includes only for the RoI head:

$$\mathcal{L}_{sup} = \sum_i \mathcal{L}_{cls}^{rpn}(\mathbf{x}_i^l, \mathbf{y}_i^l) + \mathcal{L}_{reg}^{rpn}(\mathbf{x}_i^l, \mathbf{y}_i^l) + \mathcal{L}_{cls}^{roi}(\mathbf{x}_i^l, \mathbf{y}_i^l) + \mathcal{L}_{reg}^{roi}(\mathbf{x}_i^l, \mathbf{y}_i^l), \tag{2}$$

$$\mathcal{L}_{unsup} = \sum_i \mathcal{L}_{cls}^{roi}(\tilde{\mathbf{x}}_i^u, \tilde{\mathbf{y}}_i^{pl}) + \mathcal{L}_{reg}^{roi}(\tilde{\mathbf{x}}_i^u, \tilde{\mathbf{y}}_i^{pl}). \tag{3}$$

After calculating the total loss, the student's weights are updated using the standard backpropagation algorithm. Since teacher and student models share the same network architecture, the teacher's weights $\theta_t$ are updated via an Exponential Moving Average (EMA) of the student's weights $\theta_s$:

$$\theta_t = \alpha\theta_t + (1 - \alpha)\theta_s, \tag{4}$$

where $\alpha$ is the EMA update rate. In addition to the teacher-student mechanism, Efficient-CLS incorporates an episodic memory. This memory stores a limited number of labeled images per category, which are periodically replayed during training to help mitigate catastrophic forgetting.

## 3.3 OVERVIEW OF CROPBANK

One of the key components of ACRST (Zhang et al., 2022) is a data augmentation module called CropBank. This module maintains a memory buffer that stores both labeled and pseudo-labeled object instances, where each instance corresponds to a cropped image patch extracted from the bounding box region of a single object, along with its associated category label. The CropBank memory is continuously updated with new object patches obtained from both labeled and pseudo-labeled images of the training batch. To enhance the learning process, samples from this memory are selectively pasted into unlabeled training images as additional objects. The selection strategy is category-aware: each category is assigned a sampling probability that depends on the teacher model's confidence in its predictions for that category. This design addresses two major challenges in SSOD: foreground-background imbalance (*i.e.*, background regions often dominate the training samples) and foreground-foreground imbalance (*i.e.*, unequal representation of object categories).

## 3.4 PROPOSED FRAMEWORK

In addition to replaying categories at the image level, we extend this idea to the object level. To achieve this, we build upon the Efficient-CLS (Wu et al., 2023) framework by incorporating an object memory module named *ContinualCropBank* inspired by CropBank (Zhang et al., 2022), as presented in Figure 2. For each category, this module stores a limited set of cropped object patches (bounding box regions) and their corresponding category labels. These stored instances are later pasted into labeled and unlabeled images to replay categories and mitigate catastrophic forgetting.

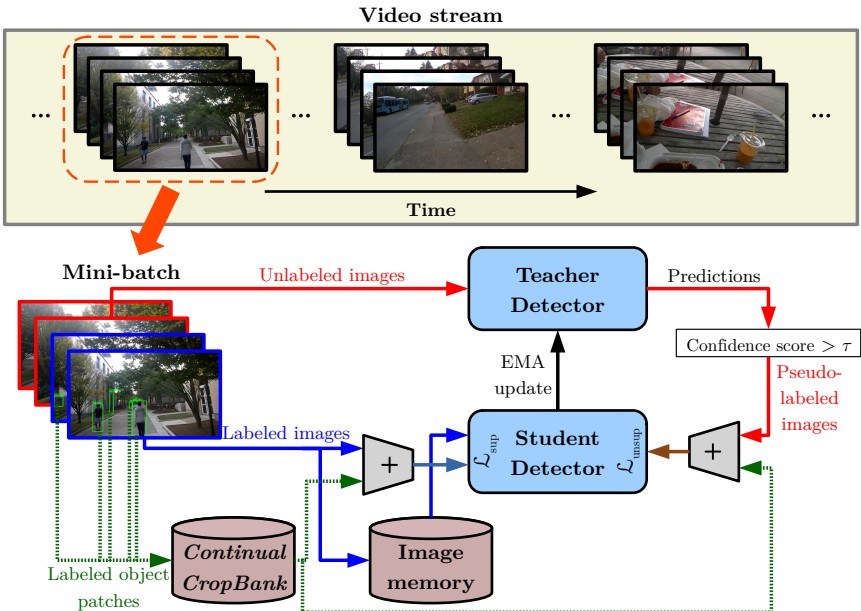

Figure 2: **The proposed ContinualCropBank integrated with Efficient-CLS** (Wu et al., 2023). The pipeline begins with a mini-batch of images streamed from video data, containing labeled and unlabeled images. Labeled images update the image memory, while their object regions are cropped to update the ContinualCropBank. The Teacher model generates pseudo-labels for unlabeled images. Then, the labeled and pseudo-labeled images are augmented with object patches from ContinualCropBank and used to train the Student model. An additional mini-batch sampled from the image memory further supports Student training. Finally, the Teacher weights are updated as the EMA of the Student weights. Image frames were extracted from the OAK dataset (Wang et al., 2021).

When a new training mini-batch becomes available after being processed by the teacher model to generate pseudo-labels, the object memory is updated with object patches extracted exclusively from labeled images. To respect the maximum capacity per category, the memory follows a first-in, first-out replacement policy. After the memory is updated, $K$ object patches are pasted into each labeled and pseudo-labeled image following these steps:

1. Randomly select a category and then randomly sample an object patch belonging to that category from the object memory;
2. Apply random scaling to the selected object patch;
3. Paste the transformed object patch into a random location in the image;
4. Repeat the above steps $K$ times.

Before pasting, we ensure that objects do not heavily occlude existing ones; otherwise, they are skipped. Random scaling is applied adaptively depending on the original object size. Mild scaling factors are used for small objects, while stronger downscaling factors are applied to medium and large objects. This strategy decreases the size of larger objects while modestly augmenting small ones, thus increasing the frequency of small and challenging objects during training, which are typically scarce in the datasets (see Table 1). Such augmentation has been shown to enhance detection

performance (Kisantal et al., 2019), and in our case, it also helps alleviate catastrophic forgetting (see Section 5). Finally, the augmented images are provided to the student model for training.

# 4 EXPERIMENTAL SETUP

## 4.1 DATASETS

We consider Objects Around Krishna (OAK) (Wang et al., 2021) and EgoObjects (Zhu et al., 2023) as benchmark datasets in our experiments. Both datasets consist of sequentially ordered image frames extracted from video streams, along with corresponding object-level annotations (*i.e.*, bounding boxes and category labels). As defined by Wang et al. (2021), the images following each mini-batch are extracted to build the test set, and the remaining images are allocated to the training set. Furthermore, we use the modified dataset versions released by Wu et al. (2023) in their public code repository[1]. These versions contain subsets of the original dataset examples, organized in temporal order, with a fixed split specifying which images are treated as labeled and which as unlabeled. The main characteristics of these modified benchmarks are summarized in Table 1.

Table 1: Characteristics of benchmark datasets used in the experiments.

| Dataset | Number of images | Number of objects | Number of classes | Image Size | Percentage of small objects |
|---------|------------------|-------------------|-------------------|------------|------------------------------|
| OAK | 30,060 | 289,728 | 103 | $648 \times 1152$ | 10.71% |
| EgoObjects | 36,457 | 89,749 | 277 | Variable | 1.53% |

## 4.2 EVALUATION METRICS

To ensure comparability with prior work (Wang et al., 2021; Wu et al., 2023), we evaluate performance in LEOCOD using three metrics: Continual Average Performance (CAP), Final Average Performance (FAP), and Forgetfulness (F). CAP measures average detection accuracy, FAP reflects detection accuracy at the end of training, and F quantifies the ability of the model to prevent catastrophic forgetting. A detailed description of these metrics is provided in the Appendix (Section A.2).

## 4.3 BASELINE MODELS

We evaluate our method against three primary baselines. **Naive Incremental Training** processes the data stream sequentially, with each example observed only once and without any mechanisms to mitigate catastrophic forgetting, serving as a lower-bound baseline. **Offline Training** assumes full access to the training dataset and allows multiple epochs of training, representing an upper-bound that is not subject to the constraints of continual or online learning. **Efficient-CLS** (Wu et al., 2023) is the current state-of-the-art approach for online continual object detection in semi-supervised settings. We combine Efficient-CLS and ContinualCropBank with several CL strategies, adapted for the object detection task. For iCaRL (Rebuffi et al., 2017), at each training iteration, the detector processes two mini-batches: one sampled from the data stream and another from the image memory for experience replay. A-GEM (Chaudhry et al., 2019) builds on iCaRL by sampling an additional smaller mini-batch from the image memory to estimate a reference gradient direction; if the gradient from the current data stream conflicts with this reference, it is projected onto the memory gradient direction. GDumb (Prabhu et al., 2020) trains the detector exclusively on two mini-batches sampled from the image memory at each iteration, using stream images only after they have been stored.

# 5 RESULTS AND DISCUSSION

Our experimental assessment comprises a comparative analysis of the proposed method against baseline approaches and the state-of-the-art method, under fully supervised and semi-supervised settings. Furthermore, we conduct ablation studies to isolate and quantify the individual contributions of the main components of our proposal.

---

[1]https://github.com/showlab/Efficient-CLS

Table 2: Experimental results under the fully supervised setting. Results marked with [†] are taken from Wu et al. (2023); all others are from our experiments. ContinualCropBank results are reported as the relative difference with respect to Efficient-CLS without our module. Best overall values in each column (excluding the offline baseline) are bold and underlined. Performance gains are shown in green, while reductions are indicated in red. ↑ indicates higher is better, ↓ indicates lower is better.

| Method | OAK | | | EgoObjects | | |
|---|---|---|---|---|---|---|
| | FAP (↑) | CAP (↑) | F (↓) | FAP (↑) | CAP (↑) | F (↓) |
| Incremental | 6.70 | 7.64 | 0.31 | 11.82 | 3.92 | 1.16 |
| Offline Training | 48.56 | 40.04 | – | 90.06 | 72.12 | – |
| EWC[†] | 7.73 | 7.02 | -0.12 | 5.15 | 1.60 | 0.57 |
| iOD[†] | 7.92 | 7.14 | -0.01 | 8.80 | 1.56 | 0.52 |
| iCaRL[†] | 22.89 | 16.60 | -2.95 | 37.61 | 21.71 | 2.79 |
| iCaRL + Efficient-CLS | 39.32 | 27.59 | -5.69 | 67.02 | 41.38 | -3.43 |
| + ContinualCropBank | **+3.23** | **+2.32** | **-2.32** | **+8.99** | +9.57 | +0.28 |
| A-GEM + Efficient-CLS | 40.42 | 27.36 | -5.75 | 66.80 | 41.00 | **-4.09** |
| + ContinualCropBank | +0.94 | +2.37 | -1.18 | +8.13 | +8.93 | +0.97 |
| GDumb + Efficient-CLS | 37.34 | 27.76 | -7.12 | 72.24 | 48.83 | -2.81 |
| + ContinualCropBank | +1.97 | -0.64 | -0.39 | +0.91 | **+2.51** | -1.26 |

**Fully Supervised Evaluation.** In this setting, we evaluate OCL for object detection under the assumption that all video frames are labeled. This scenario remains challenging because catastrophic forgetting can still degrade performance. Table 2 reports the results for this setting. As expected, the incremental baseline performs poorly due to severe forgetting, whereas the offline baseline, with access to the full dataset, achieves the strongest results. Among CL methods, EWC, iOD, and iCaRL reduce forgetting to some extent, yielding moderate gains over the incremental baseline. The results for Efficient-CLS further demonstrate that a dual-model strategy can deliver substantial improvements even under full supervision. As observed by Wu et al. (2023), these gains are likely driven by the EMA update mechanism, which stabilizes the Teacher model and mitigates forgetting.

In Table 2, results with ContinualCropBank are presented as relative improvements over Efficient-CLS without our module. When integrated with Efficient-CLS and various CL methods, Continual-CropBank enhances performance in most cases. In the few instances where performance decreases, the drop is minimal, at most 0.97 percentage points. Overall, ContinualCropBank achieves the best total results in 5 out of 6 cases, surpassing Efficient-CLS by at least 2.32 percentage points in these cases. Improvements are particularly pronounced on EgoObjects, where the performance gap to the offline baseline is larger than on OAK. For instance, ContinualCropBank improves the Efficient-CLS baseline by up to 9.57 percentage points in CAP on EgoObjects. These findings highlight that object-level replay complements image-level strategies, enabling more fine-grained retention of prior knowledge and enhancing detection performance under OCL with fully supervised conditions.

**Semi-Supervised Evaluation.** The semi-supervised experiments compare only Efficient-CLS and our method, since these are the only approaches designed for LEOCOD. In this scenario, OCL becomes even more challenging because the model must effectively incorporate unlabeled images into training. Table 3 reports the relative gains obtained by integrating ContinualCropBank into Efficient-CLS. Results for ContinualCropBank are expressed as relative performance improvements over the pure Efficient-CLS baseline. ContinualCropBank consistently improves performance across different CL methods, with the only observed decreases being marginal, at most 0.22 percentage points. In contrast, most cases show clear improvements, often exceeding 2 percentage points and reaching up to 7.21 points. Importantly, ContinualCropBank achieves the best overall results across all metrics and datasets. These findings highlight its effectiveness for LEOCOD.

To further investigate these gains, we analyze performance at varying annotation costs. Figure 3 compares Efficient-CLS and ContinualCropBank (iCaRL versions) across multiple evaluation metrics, datasets, and proportions of labeled data. Minor drops occur in only one case (the F metric at 100% on EgoObject). In general, incorporating ContinualCropBank into Efficient-CLS consistently boosts performance across metrics, datasets, and annotation costs, confirming its robustness in di-

Table 3: Experimental results under semi-supervised setting with 25% labeled samples per mini-batch. All results are from our experiments. ContinualCropBank results are reported as the relative difference with respect to Efficient-CLS without our module. Best overall values in each column (excluding the offline baseline) are bold and underlined. Performance gains are shown in green, while reductions are indicated in red. ↑ indicates higher is better, ↓ indicates lower is better.

| Method | OAK | | | EgoObjects | | |
|---|---|---|---|---|---|---|
| | FAP (↑) | CAP (↑) | F (↓) | FAP (↑) | CAP (↑) | F (↓) |
| iCaRL + Efficient-CLS | 38.30 | 26.77 | -6.56 | 62.05 | 39.44 | -3.47 |
| + ContinualCropBank | **+2.94** | **+1.56** | **-2.25** | +7.21 | +6.26 | **-0.71** |
| A-GEM + Efficient-CLS | 38.27 | 26.55 | -7.85 | 61.73 | 39.31 | -3.76 |
| + ContinualCropBank | +2.10 | +1.43 | -0.49 | +6.80 | +6.14 | +0.22 |
| GDumb + Efficient-CLS | 37.80 | 26.65 | -6.88 | 66.36 | 44.07 | -3.29 |
| + ContinualCropBank | +2.89 | +0.45 | -1.25 | **+3.73** | **+2.30** | +0.11 |

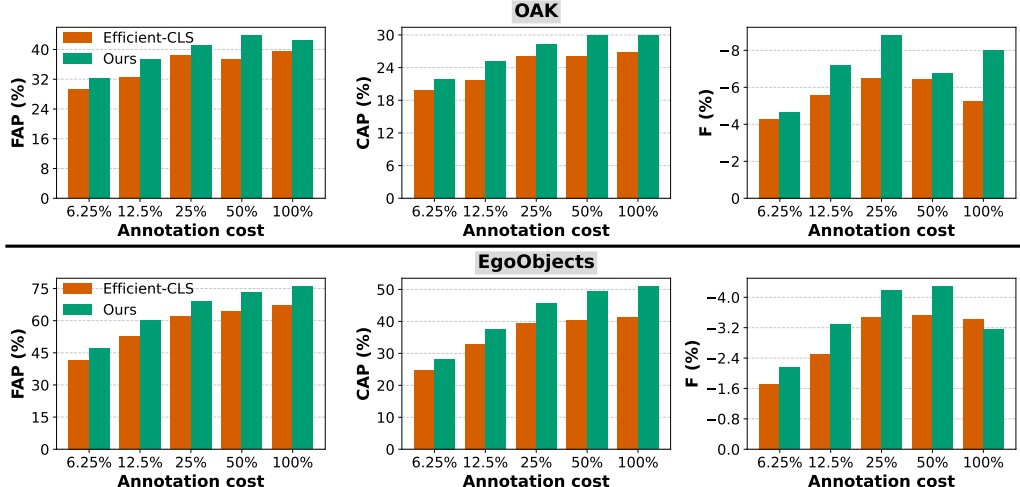

Figure 3: Detailed performance comparison of Efficient-CLS and ContinualCropBank (ours) with iCaRL across three evaluation metrics, two datasets, and varying annotation costs. Higher bars indicate better performance.

verse conditions. We also conduct a category-level analysis on three OAK classes (*tactile paving*, *potted plant*, and *sculpture*) to examine the dynamics of forgetting. Figure 4 presents detection performance for Efficient-CLS, ContinualCropBank, and the incremental baseline. For *tactile paving*, ContinualCropBank maintains stable accuracy after 1000 training steps, while Efficient-CLS experiences a sharp decline, suggesting a forgetting effect. For *potted plant*, a frequently occurring category, both methods remain stable, but ContinualCropBank consistently outperforms Efficient-CLS by a small margin. For *sculpture*, both methods exhibit late-stage drops in accuracy, yet ContinualCropBank preserves a considerably larger margin over Efficient-CLS.

## 5.1 ABLATION STUDY

We individually evaluate the components of ContinualCropBank in an incremental way, starting from the pure Efficient-CLS baseline and progressively integrating each component.

**Object Memory.** We examine the effect of integrating object-level replay into Efficient-CLS. Specifically, we extend the framework with an object memory that stores a fixed number of object patches and pastes them into labeled and pseudo-labeled images during training. This addition produces consistent improvements across all metrics and datasets, as reported in the second row of Table 4. For example, FAP and CAP increase by more than 6 points on EgoObjects, and for-

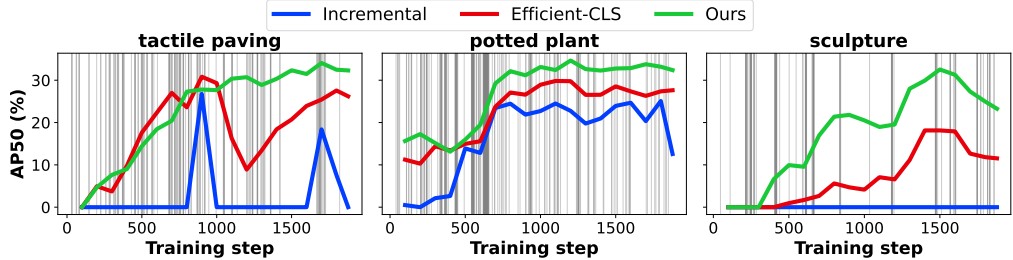

Figure 4: Progress of detection accuracy (AP50) for three categories over training with $12.5\%$ annotation cost. Vertical gray lines mark steps in which samples of the corresponding category appear.

getting reduces by $1.62$ points on OAK. These findings demonstrate the dual benefit of the object memory: boosting detection accuracy and alleviating forgetting. As object-level replay increases the density of object instances, the observed gains likely stem from addressing foreground-background imbalance. Furthermore, this component stands out as the main contributor to the improvements achieved by our method.

**Rescaling.** We further enhance the object memory by introducing random rescaling when pasting object patches into stream images, yielding the complete ContinualCropBank module. In this solution, rescaling aims to increase the representation of small objects during training. As shown in the third row of Table 4, this strategy provides additional gains in most cases. Since video frames capture objects at varying distances, robustness to scale variations, particularly for small objects, is crucial for improving detection performance. Gains are especially consistent on EgoObjects, likely due to the scarcity of small objects in this dataset. For OAK, we observe a minor drop of $0.17$ points in CAP; however, this decrease is negligible compared to the broader improvements achieved, indicating that rescaling does not compromise overall performance. Importantly, the forgetting effect decreases on both datasets by at least $0.54$ points, suggesting that the improved handling of small objects enhances model stability.

Table 4: Ablation results of ContinualCropBank components (iCaRL) on the OAK dataset with $25\%$ annotation cost. Each row incrementally adds the listed component to those above it.

| Components | OAK | | | EgoObjects | | |
|---|---|---|---|---|---|---|
| | FAP ($\uparrow$) | CAP ($\uparrow$) | F ($\downarrow$) | FAP ($\uparrow$) | CAP ($\uparrow$) | F ($\downarrow$) |
| Efficient-CLS | 38.30 | 26.77 | -6.56 | 62.05 | 39.44 | -3.47 |
| + Object Memory | 40.91 | **28.50** | -8.18 | 68.81 | 45.50 | -3.64 |
| + Rescaling | **41.24** | 28.33 | **-8.81** | **69.26** | **45.70** | **-4.18** |

## 6 CONCLUSION

In this paper, we addressed the challenge of Label-Efficient Online Continual Object Detection (LEOCOD), a setting in which models must continuously learn from temporally ordered video streams under strict storage constraints. Unlike conventional OCL approaches, each mini-batch in LEOCOD contains both labeled and unlabeled images. To address these challenges, we introduced *ContinualCropBank*, an object memory module that stores cropped patches from labeled images and reuses them for object-level augmentation during training. Beyond mitigating catastrophic forgetting through fine-grained replay, ContinualCropBank also helps alleviate foreground–background imbalance and the scarcity of small objects. Extensive experiments on two benchmark datasets demonstrate that ContinualCropBank consistently boosts detection accuracy and reduces forgetting, yielding substantial gains over Efficient-CLS and other baselines in both fully supervised and semi-supervised settings. Overall, our findings advance the state-of-the-art in LEOCOD and highlight the effectiveness of object-level replay as a simple yet powerful solution for LEOCOD.

REPRODUCIBILITY STATEMENT

To ensure reproducibility of our findings, we provide implementation details and hyperparameter settings of all methods in the Appendix (Section A.3). In addition, the source code for our ContinualCropBank module, integrated into the Efficient-CLS code[2], is available as supplementary material.

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

## A  APPENDIX

### A.1  USE OF LARGE LANGUAGE MODELS

We used a Large Language Model (LLM) as a writing assistant to improve the clarity, grammar, and flow of the manuscript. The LLM was employed solely for language polishing and did not contribute to the design of the methodology, experiments, analysis, or conclusions. All scientific content, including problem formulation, proposed approach, and empirical results, was developed entirely by the authors.

### A.2  EVALUATION METRICS DETAILS

During continual learning, the model is evaluated on the test set $T$ times at fixed intervals of $n$ training iterations. At each evaluation step $t_i$, we compute the detection performance for category $c$ using Average Precision (AP) at an Intersection over Union (IoU) threshold of 0.5, denoted $AP_{50}$. We represent this performance per category as $CAP_{t_i}^c$. The overall CAP is then defined as the mean performance across all categories and evaluation steps:

$$\text{CAP} = \frac{1}{T}\sum_{i=1}^{T} \text{CAP}_{t_i} = \frac{1}{TC}\sum_{i=1}^{T}\sum_{c=1}^{C} \text{CAP}_{t_i}^c, \tag{5}$$

where $C$ is the total number of categories. In addition, we report the FAP metric, which corresponds to $CAP_{t_i}$ at the final test step $(t_i = T)$:

$$\text{FAP} = \text{CAP}_T. \tag{6}$$

To assess the model's ability to retain previously learned knowledge, we compute the F metric. For each category $c$, the category-wise forgetfulness $F^c$ is calculated by grouping the detection scores $CAP_{t_i}^c$ according to the number of iteration steps $k$ that have elapsed between the last training occurrence of category $c$ (on labeled images) and the test step $t_i$. This process yields $K$ bins, $B_{k_{\min}}, \ldots, B_{k_{\max}}$, where each bin $B_k$ aggregates scores associated with a specific interval $k$. For each bin, we compute the average detection performance $aCAP_k$ as the mean of all $CAP_{t_i}^c$ values in the bin. This average quantifies the model's ability to detect category $c$ after $k$ training iterations without seeing any examples from that category.

The category-wise forgetfulness $F_c$ is then defined as a weighted, normalized average of performance degradation ($aCAP_{k_{\min}} - aCAP_k$), where the weights are proportional to the interval gap ($k - k_{\min}$):

$$F^c = \sum_{k=k_{\min}}^{k_{\max}} \frac{k - k_{\min}}{\sum_{k=k_{\min}}^{k_{\max}} (k - k_{\min})} \times (aCAP_{k_{\min}} - aCAP_k) \tag{7}$$

Finally, the overall forgetfulness F is calculated as the mean of $F_c$ across all categories in $C$:

$$F = \frac{1}{C} \sum_{c=1}^{C} F^c. \tag{8}$$

## A.3    IMPLEMENTATION DETAILS

In all experiments, we adopt Faster R-CNN (Ren et al., 2015) with a ResNet-50 backbone (He et al., 2016) as the detection model. The network is initialized with weights pre-trained on the PASCAL VOC dataset (Everingham et al., 2010), and the mini-batch size is fixed at 16. As the optimizer, we use adaptive moment estimation (Adam) (Kingma & Ba, 2017) with a fixed learning rate of $1 \times 10^{-4}$. For Efficient-CLS, we employ two Faster R-CNN models under the teacher-student framework. The pseudo-label confidence threshold is set to $\tau = 0.7$, the weight for the unsupervised loss term (Equation 1) is $\lambda = 1.0$, and the EMA update rate for teacher parameters (Equation 4) is fixed at $\alpha = 0.99$. The image memory module stores a maximum of 5 images for each category. For the ContinualCropBank module, it stores a maximum number of 20 object patches per category. Scaling factors ranging from 1.1 to 1.2 are applied to small objects, and scaling factors ranging from 0.4 to 0.9 are used for medium and large objects.

For iCaRL, the mini-batches of the current data stream and image memory contain 16 images each. For A-GEM, 4 images are drawn from the memory to estimate the gradient direction. For GDumb, both mini-batches sampled from the image memory have 16 images. The offline training baseline runs for 5 epochs over the training set, with images shuffled in random order. Finally, training is performed on two NVIDIA RTX A5000 GPUs in parallel and testing occurs after every 100 iterations using the same set on all models. Our implementation is built upon the official Efficient-CLS implementation, publicly available at its repository[3].

---

[3]https://github.com/showlab/Efficient-CLS

