# OpenReview forum: "ContinualCropBank: Object-Level Replay for Semi-Supervised Online Continual Object Detection"
_ICLR.cc/2026/Conference — ICLR 2026 Conference Withdrawn Submission_

### Official Review · Reviewer_DPFm · 2025-10-24

**Soundness:** 3
**Presentation:** 3
**Contribution:** 3
**Rating:** 6
**Confidence:** 4

**Summary:**

The paper addresses the problem of supervised and semi-supervised online continual learning for object detection, and proposes a novel replay module, ContinualCropBank, inspired by CropBank, initially developed for SSOD. The module is based on the idea of storing object patches cropped from the bounding-box regions of labeled images, and pasting them on streaming images during training. The module can be used on top of other modules such as Efficient-CLS, and they can both be used on top of other methods. Methods from the literature combined with these modules are tested in a fully supervised and a semi-supervised scenario, and exhibit improvements with respect to the standalone methods.

**Strengths:**

The authors present a novel module that draws inspiration from the online continual learning literature and adapts it to the semi-supervised setting of LEOCOD. Although the concept of CropBank was previously introduced in the literature, its adaptation to the LEOCOD framework, particularly when combined with Efficient-CLS, is non-trivial. The proposed module demonstrates notable improvements over standalone methods, as well as over methods combined with Efficient-CLS alone. The ablation study further supports the effectiveness of the module and strengthens the overall evaluation.

**Weaknesses:**

My main concern is on whether the comparison with previous approaches is appropriate. Since the proposed method is suitable for Semi-supervised Online Continual Learning for Object Detection, it should be compared against methods designed for subareas of this problem such as:
- Continual Learning for object detection
- Online Continual Learning for Object Detection
- Semi-supervised Continual Learning for object detection
Indeed, a method from the Semi-supervised Continual Learning for object detection literature could have a good performance (maybe analogous to the proposed method) even if it isn’t strictly created for Online CL settings. Similarly, methods for Online Continual Learning for Object Detection should be tested in the “fully supervised” subsection of the experiments. Instead, the authors only compare against methods from the general continual learning literature, hence the improvements that they report could be misleading.
I would suggest adding comparisons with such methods, or better justifying the lack of such comparison.

Minor concerns:
Lines 053-065 are difficult to follow, especially for how LEOCOD, online continual learning, SSOD, efficient-CLS are tied together. Table 2 is a little bit confusionary, I would like to see also the improvement of Efficient-CLS alone.

**Questions:**

Besides the concerns raised in the weaknesses section, here are additional questions:
- In Table 2 we can notice that there is a great improvement regarding methods paired with efficient-cls and methods alone. From my understanding, however, efficient-cls was created for the semi-supervised settings. Why is the improvement this strong in a fully supervised setting?
- In table 2, sometimes some metrics improve thanks to ContinualCropBank, while others decrease. Why do you think that there is this mismatch sometimes? Why is forgetting the metric that decreases most often?

---

> ### Author Response · Authors · 2025-11-21
>
> Thank you for the very thoughtful comments on baselines, positioning, and metrics; they help clarify how our work fits within continual and semi-supervised detection, and how the reported gains should be interpreted.
>
> Our focus is the LEOCOD setting, which combines online continual learning (one-pass, temporally ordered video stream) with semi-supervised detection (mixed labeled and unlabeled images per mini-batch). Efficient-CLS was specifically proposed for this protocol, integrating a teacher–student SSOD scheme and image-level replay, and is currently the strongest published method in this exact setting, so our main empirical comparisons are built around it plus standard CL baselines adapted to detection. Many prior continual or semi-supervised continual detectors operate in task-based, multi-epoch, or fully supervised streams and do not satisfy all LEOCOD constraints simultaneously, so porting them fairly would require non-trivial changes rather than a drop-in comparison. In a future version of the paper, we plan to clearly categorize these subareas, justify which methods cannot be straightforwardly adapted, and, where feasible, add further comparisons or discussion-style positioning to avoid misleading impressions.
>
> LEOCOD can be read as “OCL + SSOD”: it inherits the one-pass, order-respecting stream from OCL and the labeled/unlabeled split and teacher–student pseudo-labeling from SSOD. Efficient-CLS instantiates this combination, and ContinualCropBank then adds object-level replay on top of Efficient-CLS to better address catastrophic forgetting, foreground-background imbalance, and small-object scarcity. In a future version, we plan to rewrite lines 53–65 to spell out this hierarchy more clearly and update Table 2 to explicitly show both (i) the improvement of Efficient-CLS over Incremental/CL baselines and (ii) the additional gains from adding ContinualCropBank.​​
>
> Regarding Table 2, Efficient-CLS still helps in the fully supervised case because, even without unlabeled images, it retains key components such as the EMA teacher and episodic image replay, which stabilize training and mitigate forgetting in the online stream. This explains the strong gains over generic CL methods that were originally developed for classification or task-incremental settings. ContinualCropBank then complements Efficient-CLS by adding object-level replay, yielding further improvements in most cases and only very small drops when they do occur.
>
> FAP, CAP, and forgetting F capture different aspects of performance: FAP is average performance over time, CAP is final performance, and F measures how much earlier performance degrades after seeing later data. An explanation for occasional drops in the forgetting metric F is that ContinualCropBank can inflate the peak performance for some classes earlier in training more than it improves their final performance, which mathematically increases the gap that F measures. Because object-level replay injects additional, often easier instances (e.g., less cluttered or rescaled copies) early on, some classes can reach a higher maximum AP during the stream; if their final AP improves only slightly or remains similar, the difference “max AP − final AP” grows, and F becomes worse even though CAP and FAP are equal or better. In other words, the module sometimes over-boosts the intermediate peak for certain classes relative to the end-of-stream performance, leading to higher measured forgetting for those classes despite overall accuracy gains. In a future version of the paper, this effect will be clarified and, if feasible, illustrated with temporal per-class AP curves.

---

### Official Review · Reviewer_oKGc · 2025-10-28

**Soundness:** 2
**Presentation:** 2
**Contribution:** 1
**Rating:** 2
**Confidence:** 4

**Summary:**

The paper studies Label-Efficient Online Continual Object Detection, an online one-pass continual detection setting where each mini-batch contains only a small number of labeled images and the rest are unlabeled images, and the model is trained using a teacher–student framework similar to Efficient-CLS. The proposed module, ContinualCropBank, stores ground-truth object crops in a per-class FIFO memory and pastes these rescaled crops back into current labeled and unlabeled images to increase foreground density and expose more small and rare objects.

**Strengths:**

LEOCOD is realistic and under-explored: it is online, single-pass, and only partially labeled at each step. ContinualCropBank is simple to integrate into Efficient-CLS and is reported to improve CAP and FAP and reduce Forgetfulness F under different supervision ratios.

**Weaknesses:**

1.	Novelty is not clearly demonstrated. ContinualCropBank is conceptually very close to prior instance-bank and copy-paste style object replay methods in semi-supervised object detection. The paper claims that per-class FIFO storage, using only ground-truth crops instead of pseudo-label crops, and explicitly targeting catastrophic forgetting are the main contributions. However, the paper does not include a baseline that directly applies a naïve CropBank-style replay strategy under the same one-pass, partially labeled LEOCOD protocol. This baseline should be added to show that existing object-level replay does not already solve this setting.
2.	The paper does not disentangle continual replay from stronger augmentation. ContinualCropBank pastes rescaled crops into current training images, injecting extra foreground content and creating additional small objects, which by itself can improve object detection performance regardless of solving catastrophic forgetting. A baseline should be included to verify that the gains in CAP and FAP and the reductions in Forgetfulness F actually come from mitigating catastrophic forgetting rather than simply from aggressive data augmentation.
3.	Technical details important for reproducibility are under-specified. The paper does not precisely define how severe occlusion is avoided when pasting crops, how small, medium, and large objects are defined for the rescaling policy, or how sensitive performance is to the fixed per-class FIFO capacity of stored crops. These thresholds, definitions, and sensitivity analyses should be stated explicitly.

**Questions:**

As listed above:
1. What is the novelty of the proposed method?
2. How to prove the gains of the paper come from the proposed modules?

---

> ### Author Response · Authors · 2025-11-21
>
> Thank you for the constructive feedback, which helps clarify both the novelty and the source of the gains of ContinualCropBank in the LEOCOD setting.
>
> ContinualCropBank is designed specifically for LEOCOD, where data arrive as a one-pass video stream with mixed labeled and unlabeled images and a strict memory budget, unlike prior instance-bank methods (e.g., CropBank) developed for offline SSOD with multiple epochs and effectively unbounded replay. Concretely, it (i) stores only ground-truth crops from labeled images, (ii) uses per-class FIFO capacity to keep a balanced and temporally updated object memory, and (iii) integrates this memory into Efficient-CLS’s teacher–student pipeline to provide object-level replay under continual constraints. This combination (labeled-only per-class FIFO replay, scale-aware pasting, and explicit targeting of catastrophic forgetting in a one-pass semi-supervised continual setting) is not covered by existing copy–paste instance banks designed for static datasets.
>
> The ablation in the paper progressively adds components on top of Efficient-CLS: baseline, +object memory, and +object memory +rescaling. Adding only the object memory already yields large gains in FAP and CAP and reductions in forgetting F on both OAK and EgoObjects, showing that temporal object replay itself substantially improves retention and accuracy, beyond generic augmentation. Adding rescaling then brings smaller, consistent extra gains, indicating that scale-aware augmentation is complementary rather than the primary driver.​
>
> Furthermore, temporal per-class AP curves show that classes with long gaps (e.g., “tactile paving”, “sculpture”) maintain higher performance over time with ContinualCropBank than with Efficient-CLS, consistent with reduced catastrophic forgetting rather than merely stronger single-batch augmentation. These trends hold across datasets, label budgets, and multiple continual learners (iCaRL, A-GEM, GDumb), suggesting that the improvements in CAP, FAP, and F stem from the proposed replay design rather than from a fragile augmentation trick.
>
> In a future version of the paper, we plan to add two complementary baselines: (i) a naïve CropBank-style replay adapted to LEOCOD (storing labeled and pseudo-labeled crops without per-class FIFO), and (ii) a “strong augmentation without replay” baseline (copy–paste from the current mini-batch only), to more cleanly isolate object-level replay across time from purely within-batch augmentation.

---

### Official Review · Reviewer_92aK · 2025-10-29

**Soundness:** 2
**Presentation:** 2
**Contribution:** 2
**Rating:** 2
**Confidence:** 5

**Summary:**

The paper targets label-efficient online continual object detection. Building on Efficient-CLS, it integrates the CropBank technique from semi-supervised object detection, augments it with FIFO and per-class capacity limits, and applies scaling and related data-augmentation strategies, yielding the proposed ContinualCropBank method. Compared with classic baselines (e.g., EWC) and the domain baseline Efficient-CLS, the approach achieves substantial performance improvements.

**Strengths:**

By integrating and adapting existing methods to the streaming-data setting, this paper improves upon prior LEOCOD approaches and offers a clear design rationale supported by strong empirical evidence.

**Weaknesses:**

1.The paper stacks the CropBank component from ACRST (semi-supervised object detection) onto the prior LEOCOD method Efficient-CLS and adds FIFO, per-class quotas, and data augmentation. This is essentially a scenario-specific adaptation rather than a genuine innovation.
2.Random placement, fixed scaling ranges, and basic occlusion checks may introduce artificial artifacts, leading to unrealistic context/occlusion patterns and texture repetition.
3.Compared with CropBank, the proposed ContinualCropBank stores patches only from labeled images and discards pseudo-labeled ones, but the rationale for this design choice is not explained.
4.The paper does not analyze sensitivity to key hyperparameters (number of pastes per image K, scaling ranges, memory capacity, IoU/occlusion thresholds) nor justify why FIFO is preferable to alternative policies.

**Questions:**

1.Please clarify how the proposed ContinualCropBank differs from CropBank. In particular, if CropBank were applied directly to the LEOCOD setting, would its results differ from those reported for ContinualCropBank?
2.Please justify—theoretically or empirically—the advantage of extracting patches only from labeled images (i.e., discarding pseudo-labeled images) compared with the original CropBank design.
3.Please demonstrate the sensitivity of key hyperparameters (e.g., number of pastes per image K, scaling ranges, memory capacity, IoU/occlusion thresholds) to support the choices made in the paper.
4.Please provide runtime, memory usage, and throughput overhead to quantify the additional computational cost introduced by the method.

---

> ### Author Response · Authors · 2025-11-21
>
> Thank you for the detailed and constructive comments, which help clarify how ContinualCropBank relates to prior SSOD work (especially CropBank) and to Efficient-CLS in the LEOCOD setting, as well as highlight open points about design choices, hyperparameters, and computational overhead.
>
> CropBank was proposed for offline semi-supervised detection on static datasets, where the model can iterate over the full labeled and unlabeled sets for multiple epochs, and catastrophic forgetting is not a concern. Its memory stores instance crops from both ground-truth and pseudo-labeled boxes, and when sampling crops, it uses an adaptive class-rebalancing distribution based on Pseudo Recall PR_k, explicitly up-weighting hard or under-recalled classes and down-weighting confident / frequent ones to correct foreground–foreground and foreground–background imbalance. ContinualCropBank, in contrast, is designed for LEOCOD, where video frames arrive in temporal order, are seen once, and a strict memory budget is enforced, so it (i) stores only crops from labeled images, (ii) maintains a bounded per-class pool of object patches with FIFO replacement, and (iii) samples approximately uniformly within each class while integrating this replay into Efficient-CLS’s teacher–student pipeline to mitigate forgetting under online constraints. In a one-pass LEOCOD stream, pseudo-label quality and class confidence evolve over time, so directly transplanting CropBank (i.e., storing pseudo-labeled crops and adapting sampling via PR_k computed from a non-stationary pseudo-label set) would make replay tightly coupled to changing, potentially noisy confidence estimates, without any constraint on how long such crops remain in memory. By contrast, ContinualCropBank’s labeled-only, per-class FIFO memory is explicitly designed for this setting to keep the replay buffer cleaner, class-balanced, and more temporally representative, which is better suited for mitigating forgetting under strict one-pass and memory constraints.​​ In a future version of the paper, we will make this distinction between CropBank and ContinualCropBank in LEOCOD more explicit and, where possible, add experiments to empirically support these arguments.
>
> In offline ACRST, pseudo-label quality can be refined over multiple epochs, making it safer to store pseudo-labeled crops in memory. In LEOCOD, pseudo-labels, especially early ones, can be significantly noisier. If stored, their errors would be replayed repeatedly into future images, turning transient mistakes into persistent biases in the buffer. By restricting storage to labeled images, ContinualCropBank ensures the object memory is populated only with ground-truth boxes, providing a high-precision replay source while still pasting those patches into both labeled and pseudo-labeled images during training. This design prioritizes robustness to error propagation in the one-pass, semi-supervised continual setting. In a future version of the paper, we plan to state this rationale explicitly and, where possible, support it with additional experiments.
>
> The main hyperparameters are the number of pastes per image, K, scaling ranges, memory capacity (with per-class quotas), and occlusion thresholds. In the current work, we choose conservative values based on prior SSOD augmentation practice and preliminary trials and then keep them fixed across all datasets, budgets, and continual learners. The fact that ContinualCropBank consistently improves over Efficient-CLS across OAK and EgoObjects and across multiple CL strategies suggests the method is reasonably robust within these ranges, and the ablation indicates the primary gains come from object-level replay itself, with rescaling providing additional, though smaller improvements. In a future version, we plan to include a dedicated sensitivity study varying K, capacity, scaling ranges, and occlusion thresholds to better justify these choices and provide guidance for practitioners.
>
> ContinualCropBank adds (i) a bounded object-memory buffer, controlled via per-class quotas and FIFO, and (ii) lightweight crop–paste operations plus simple geometric checks at training time. The detector backbone and heads are unchanged, so GPU FLOPs remain dominated by Efficient-CLS, and the additional cost is mainly CPU-side preprocessing and a modest increase in effective foreground density per image. In our implementation, training remains practical on the same hardware as Efficient-CLS, but we have not yet reported detailed profiling numbers. In a future version of the paper, we plan to include a small table comparing runtime and GPU memory for Efficient-CLS with and without ContinualCropBank.

---

### Official Review · Reviewer_r1QX · 2025-11-05

**Soundness:** 2
**Presentation:** 3
**Contribution:** 2
**Rating:** 2
**Confidence:** 5

**Summary:**

The paper introduces ContinualCropBank, an object-level replay module for label-efficient online continual object detection. It stores cropped object patches from labeled frames and pastes them into new labeled or unlabeled images during training, providing fine-grained replay that reduces forgetting and improves detection of small or rare objects. The experiments are evaluated on two benchmarks: OAK and the EgoObjects datasets.

**Strengths:**

- The paper addresses an important challenge in label-efficient online continual learning, where object annotations may be unavailable during training.
- The proposed approach introduces a simple yet effective object-level replay mechanism that is both memory-efficient and easily integrable with existing frameworks.
- The paper is well written and easy to follow.

**Weaknesses:**

- Conceptually inconsistent: The replay mechanism pastes objects randomly without considering the scene or spatial context, which can lead to unrealistic or semantically implausible compositions. This randomness may introduce anti-causal or spurious correlations in the semi-supervised setting; for example, placing a laptop on the roof of a moving car, causing the model to learn associations that rarely occur in real-world data. While such augmentation may improve accuracy, the gains could partly arise from exploiting these spurious correlations rather than genuine generalization.

- Weaker experimentations and missing relevant works: Baseline comparisons are limited to older continual learning methods; stronger memory-based or transformer-based COD baselines are missing. For instance, transformer-based: OW-DETR [1], transformer-memory-based: MD-DETR [2].

- The method assumes accurate bounding boxes for labeled samples, making it less robust to noisy or weak annotations.

[1] Gupta, Akshita, et al. "Ow-detr: Open-world detection transformer." Proceedings of the IEEE/CVF conference on computer vision and pattern recognition. 2022

[2] Bhatt, Gaurav, James Ross, and Leonid Sigal. "Preventing catastrophic forgetting through memory networks in continuous detection." European Conference on Computer Vision. Cham: Springer Nature Switzerland, 2024.

**Questions:**

- MD-DETR [1] generates pseudo labels through background relegation while training on current tasks, effectively functioning as a form of semi-supervised continual detection. Could the authors clarify how their approach compares to or differs from MD-DETR in terms of the semi-supervised strategy it employs?
- Is there any relationship between the selected object patch and the target image where it is pasted? How does random object insertion compare to context-aware pasting in terms of performance and realism?
- Could the authors include comparisons with more recent transformer-based and memory-augmented transformer baselines, such as [1] and [2]?


[1] Bhatt, Gaurav, James Ross, and Leonid Sigal. "Preventing catastrophic forgetting through memory networks in continuous detection." European Conference on Computer Vision. Cham: Springer Nature Switzerland, 2024.

[2] Gupta, Akshita, et al. "Ow-detr: Open-world detection transformer." Proceedings of the IEEE/CVF conference on computer vision and pattern recognition. 2022

---

> ### Author Response · Authors · 2025-11-21
>
> We thank the reviewer for the insightful and constructive feedback, which has helped us clarify the scope of our setting, the design choices behind our replay mechanism, and how our work relates to recent transformer-based continual detection methods.​​
>
> The focus of our paper is the Label-Efficient Online Continual Object Detection (LEOCOD) setting. The cited transformer-based methods are designed for task- or class-incremental continual detection on COCO and VOC with fully labeled tasks and multi-epoch training, targeting open-world detection (OW-DETR) and background relegation (MD-DETR), rather than label-efficient, one-pass video streams with mixed labeled and unlabeled data. Adapting OW-DETR or MD-DETR into fair LEOCOD baselines would thus require substantial re-engineering (e.g., explicit unlabeled data handling, a teacher–student scheme, strict one-pass streaming, and memory-budget alignment) rather than a straightforward reuse. Given this mismatch and the scope of the current work, we empirically focus on improving the state-of-the-art LEOCOD baseline Efficient-CLS. In a future version of the paper, we plan to position ContinualCropBank relative to OW-DETR and MD-DETR in the related work section.
>
> Regarding the quality of labeled bounding boxes, ContinualCropBank does not assume that annotations are perfectly accurate, but it does rely on the same labeled subset and box annotations as the underlying LEOCOD setting and Efficient-CLS baseline. Because patches are relatively local crops and we enforce basic size and non-occlusion constraints when pasting, moderate box imprecision mainly affects some background pixels within the patch and is unlikely to create severe label noise, especially given that the student also sees the original labeled images and teacher-generated pseudo labels. Importantly, any systematic bias in the labeled boxes would also affect Efficient-CLS and other LEOCOD methods, so ContinualCropBank is not uniquely vulnerable. In fact, restricting the memory to labeled crops avoids amplifying additional noise from pseudo labels.
>
> Currently, patches are sampled from the object memory without explicit semantic matching to the target image: instances are drawn approximately uniformly per category and pasted at random spatial locations, subject to non-occlusion checks and scale constraints on object size. Despite this scene-agnostic design, object-level replay with rescaling yields substantial gains over Efficient-CLS in both fully supervised and semi-supervised LEOCOD, improving average detection performance and reducing forgetting on both OAK and EgoObjects. These findings are consistent with prior SSOD work (e.g., CropBank and MixPL), where copy–paste and mosaic augmentations, even when not perfectly realistic, improve generalization under label scarcity by mitigating foreground-background imbalance and small-object scarcity. We acknowledge that random pasting can create visually implausible scenes, but note that evaluation is always performed on the original, unaugmented video streams; if the model were mainly exploiting spurious correlations (e.g., “laptop on car roof”), we would expect a drop in test performance rather than the consistent improvements observed across datasets, budgets, and continual-learning backbones. Furthermore, the fraction of pasted objects per image is small and pasting is constrained by size and occlusion checks, so the original scene structure remains dominant, similar to the design of existing SSOD copy–paste strategies that have been empirically validated. We agree that context-aware pasting could further improve realism and potentially performance, but it would add complexity and runtime in an already constrained online streaming setting.​ In a future version of the paper, we plan to explicitly state that our current scene-agnostic design is a deliberate trade-off, justified by the observed gains, and to highlight context-aware object selection and placement as an orthogonal direction for future work.

---

### Note · Authors · 2025-11-21

**Comment:**

Thank you again to all reviewers for the constructive and detailed feedback on our submission. After carefully considering the review, we accept that, while the committee found the direction promising, our current version does not yet provide sufficient experimental support and baseline coverage to convincingly demonstrate the contribution, especially regarding comparisons and additional analyses. We have therefore decided to withdraw the paper from this venue and will use the reviewers’ input to substantially extend the experiments, clarify the positioning with respect to related work, and better justify our design choices in a future version.

**Withdrawal Confirmation:**

I have read and agree with the venue's withdrawal policy on behalf of myself and my co-authors.